# Implications of Pharmacokinetic Potentials of Pioglitazone Enantiomers in Rat Plasma Mediated through Glucose Uptake Assay

**DOI:** 10.3390/molecules28134911

**Published:** 2023-06-22

**Authors:** Tatineni Spandana, Veera Venkata Nishanth Goli, Mohamed Rahamathulla, Sirajunisa Talath, Riyaz Ali M. Osmani, Mohammed Muqtader Ahmed, Syeda Ayesha Farhana, Shalam Mohamed Hussain, Bannimath Gurupadayya

**Affiliations:** 1Department of Pharmaceutical Chemistry, JSS Academy of Higher Education & Research, Mysore 570015, India; spandanatatineni4@gmail.com (T.S.); nishanthvice@gmail.com (V.V.N.G.); 2Department of Pharmaceutics, College of Pharmacy, King Khalid University, Abha 61421, Saudi Arabia; 3Department of Pharmaceutical Chemistry, RAK College of Pharmaceutical Sciences, RAK Medical and Health Sciences University, Ras Al Khaimah 11172, United Arab Emirates; sirajunisa@rakmhsu.ac.ae; 4Department of Pharmaceutics, JSS College of Pharmacy, JSS University, Mysuru 570006, India; riyazosmani@gmail.com; 5Department of Pharmaceutics, College of Pharmacy, Prince Sattam Bin Abdul Aziz University, Al Kharj 11942, Saudi Arabia; muqtadernano@gmail.com; 6Department of Pharmaceutics, Unaizah College of Pharmacy, Qassim University, Unaizah 51911, Saudi Arabia; a.farhana@qu.edu.sa; 7Department of Clinical Pharmacy, College of Nursing and Health Sciences, Al-Rayyan Medical College, Madinah 20012, Saudi Arabia; sm.hussain@amc.edu.sa

**Keywords:** pioglitazone enantiomers, enantiomeric separation, validation, non-radioactive glucose uptake assay, stereoselectivity, pharmacokinetics

## Abstract

Pioglitazone, a PPAR-gamma activator used to diagnose hyperglycemia, was studied for its stereoselective deposition and active enantiomers in female albino Wistar rats. In accordance with USFDA recommendations, a bioanalytical technique was employed to assess the segregation of pioglitazone enantiomers in rat plasma with glimepiride as an internal standard. A Phenomenox i-Amylose-3 column (150 mm × 4.6 mm) of 5 µm was used for high-performance liquid chromatography (HPLC) with a mobile phase of 10 mM ammonium acetate buffer in Millipore water and acetonitrile in 60:40 (*v*/*v*) admixture with column temperature 35 °C, wavelength 265 nm, and flow rate 0.6 mL/min, respectively. Pioglitazone-S, Pioglitazone-R, and the internal standard had retention times of 3.1, 7.4, and 1.7 min, respectively. The study found that within-run and between-run precision ranged from 0.1606–0.9889% for Pioglitazone-R and from 0.2080–0.7919% for Pioglitazone-S, while the accuracy ranged from 99.86 to 100.36% for Pioglitazone-R and 99.84 to 99.94% for Pioglitazone-S. In addition, a non-radioactive glucose uptake assay was employed to examine the enantiomers in 3T3-L1 cell lines by flow cytometry. Significant differences were demonstrated in Cmax, AUClast (h*μg/mL), AUCINF obs (h*μg/mL), and AUC%Extrap obs (%) of Pioglitazone-R and S in female albino Wistar rats, suggesting enantioselectivity of pioglitazone.

## 1. Introduction

Hyperglycemia is a persistent condition characterized by the body’s use of glucose (blood sugar) as fuel. Unlike type 1 diabetes, which is caused by a deficiency of insulin, type 2 diabetes is often the result of insulin resistance, which means the body’s cells become less responsive to insulin’s signals to take up glucose from the bloodstream. It induces elevated blood glucose levels, resulting in various consequences, including coronary heart disease, nerve failure, renal dysfunction, and visual impairment. The prevalence of type 2 diabetes has been steadily increasing worldwide, primarily due to sedentary lifestyles, unhealthy diets, and obesity. Anti-diabetic medications such as pioglitazone (PIO) can treat these disorders [1]. Chirality refers to molecules with non-superimposable mirror images. In other words, chiral molecules exist in two forms, mimicking each other. However, like hands, they cannot be superimposed over each other. Many drugs are chiral, meaning they exist in two different enantiomers which are mirror-image isomers of the same molecule. PIO is a chiral drug that displays a wide spectrum of bioavailability, distribution, and pharmacodynamic characteristics from a pharmacological and toxicological perspective [2]. The problem of chirality plagues modern medications. The human body can often distinguish between the enantiomers of a drug, even though they have the same chemical formula and molecular weight. This is because the body contains enzymes and receptors that are themselves chiral and can interact selectively with one enantiomer but not the other. This can lead to differences in the pharmacokinetics and pharmacodynamics of a drug, as well as its potential for toxicity and side effects. In this context, PIO is a chiral drug that is used to treat type 2 diabetes. The two enantiomers of PIO have different pharmacokinetic profiles, meaning that they are absorbed, distributed, metabolized, and eliminated differently by the body. The body reacts differently to each racemic substance, using a chiral selection mechanism to generate an array of biological features. Consequently, to optimize the therapeutic efficacy of chiral drugs, it may be necessary to separate the two enantiomers and administer only the active isomer. This process is known as enantiomeric enrichment or chiral switching. Alternatively, the two enantiomers can be administered together as a racemic mixture, but this can sometimes lead to undesirable side effects due to differences in their pharmacokinetics and pharmacodynamics [3].

Pioglitazone(PIO) belongs to a class of drugs known as PPAR-gamma activators, which work by activating a receptor in the body that regulates glucose and lipid metabolism. When used as a monotherapy, PIO can improve insulin sensitivity and reduce glucose production in the liver, resulting in lower blood glucose levels. It can also improve lipid profiles and reduce cardiovascular complication risk in people with type 2 diabetes. PIO is often used in combination with other medications such as sulfonylurea, metformin, or insulin to help achieve optimal glycemic control in people with type 2 diabetes who have not been able to achieve adequate blood glucose control with monotherapy alone [4]. The structure of PIO, with a carbonyl group attached to the thiazolidinedione ring, is depicted in Figure 1.

In vitro assays are frequently employed as preliminary screening techniques to assess the antidiabetic efficacy of drugs, enabling screening of a wide range of potential therapy contenders. The glucose uptake assay on 3T3L1 cell lines was evaluated using flow cytometry to ascertain its antidiabetic efficacy [5]. Glucose uptake within adipocytes and insulin’s role in sustaining glucose homeostasis can be studied using adipocytes. 3T3-L1 fibroblasts, which develop into adipocytes during adipogenesis, have been the subject of a comprehensive study on adipogenesis [6]. Diabetes studies involving glucose uptake by cells show that adipocytes are more responsive to insulin after differentiation than other cells. In response to insulin response, glucose transporter-4 (GLUT4) moves from the intracellular region (localized within vesicles) to the plasma membrane. GLUT4 serves a critical function in maintaining the body’s glucose homeostasis. Consequently, when blood glucose levels are lowered, mature adipocytes can acquire high glucose levels even at lower insulin concentrations [7]. The fluorescent glucose analog 2-[N-(7-nitrobenz-2-oxa-1,3-diazol-4-yl) amino]-2-deoxy-d-glucose (2-NBDG), which is transported within cells by GLUT4, the same transporter that regulates glucose transport within cells, has the potential to assess glucose uptake in cells by a non-invasive, simple, and safe method [8].

In this study, we aim to establish a simpler, more specific, and more accurate technique for distinguishing between the two enantiomers of PIO in rat plasma. Accurately identifying and quantifying Pioglitazone-R (PIO-R) and Pioglitazone-S (PIO-S) is essential for evaluating drug safety and efficacy. This is because enantiomers can have significantly different pharmacological properties, such as potency and toxicity, even though they have the same chemical formula [9]. Previous research has mostly used chiral high-performance liquid chromatography (HPLC) methods to differentiate between PIO enantiomers [10]. Our ultimate goal is to determine which PIO enantiomer exhibits significantly enhanced activity over the other. Specifically, we will compare the activity of the dextrorotatory (R) and levorotatory (S) enantiomers to assess which one is more effective. Therefore, this study examined the pharmacokinetic differences between PIO enantiomers in albino Wistar female rats, since hepatic glucose production is faster in the female cerebellum, heart, and brain than in male diabetic rats [11]. Our findings will help improve our understanding of the pharmacological effects of PIO enantiomers and potentially lead to the development of more effective and safer drugs.

## 2. Results

### 2.1. Implications of Assay Conditions on Segregation

Reverse-phase liquid chromatography was performed using a Lux I-Amylose-3 column with dimensions of 150 mm × 4.6 mm and a 5 µm particle size to separate PIO enantiomers and an internal standard in rat plasma. Investigating the mobile phase’s composition, the column’s flow rate, and the temperature composition led to the discovery of the ideal separation parameters. To separate PIO chiral compounds, either ammonium bicarbonate buffers (pH 7.8) and acetonitrile in varied ratios or potassium dihydrogen phosphate buffers (pH 7.0) and acetonitrile were employed, employing reversed-phase columns such as ACI Cellu and Chiralpak AD-RH as stationary phases. The low resolution may be due to the lack of stereotypical activity from the analyte or to weak enantiomer affinity for the CSP. Normal phase separation was also attempted using dissimilar proportions of n-hexane:2-propanol in the mobile phase, such as 70:30 *v*/*v* and 60:40 *v*/*v*, but failed. The effectiveness of three different chiral columns, chiralcel OD-H, chiral ART Amylose C-Neo, and chiralpak IA, was investigated with and without DEA in the mobile phase. However, the results showed poor peak shapes and longer run times, indicating insufficient separation. Although peak splitting was observed, it could not be effectively resolved using mobile phases such as 0.1% formic acid in Millipore water and methanol (20:80 *v*/*v*) or 10 mM ammonium acetate buffer and methanol (70:30 *v/v*) with a Phenomenox i-amylose-3 (150 mm × 4.6 mm) 5 µm column. This could be due to the lack of interaction between the column packing material and the mobile phase. In addition, the mobile phase cannot effectively separate chiral peaks. In this optimized method, a mobile phase comprising of acetonitrile and ammonium acetate buffer was used to achieve high resolution between PIO enantiomers using a Phenomenex i-Amylose-3 column. The column temperature, flow rate, and mobile phase proportions were adjusted to achieve the desired separation. A flow rate of 0.6 mL/min and a temperature of 35 °C were chosen for the column. The elution was monitored at 265 nm, and a resolution exceeding 6.0 was obtained between the two isomers (Figure 2b).

### 2.2. Specificity and Chromatography

The peaks of I.S. and PIO-S, PIO-R were discernible in rat plasma under the ideal conditions mentioned above, with retention periods of 1.7, 3.1, and 7.4 min, respectively. By introducing individual standards of PIO-R and PIO-S into HPLC, the sequence of the enantiomer’s elution was proven. Typical chromatograms of blank rat plasma, PIO racemate standard solution, and the I.S., which were spiked with blank plasma, Individual PIO-R and PIO-S standard solution, and plasma samples collected at 4 and 48 h following an oral PIO racemate dose of 30 mg/kg where no interfering peaks were identified are displayed in Figure 2. Results indicated that the method had high specificity and excellent selectivity.

### 2.3. Validation of Assay

#### 2.3.1. Linearity and Calibration Curve

Separated rat plasma was mixed with standard solutions containing 10–1000 µg/mL PIO and 10 µg/mL IS (Glimepiride). The calibration curve was obtained for the linearity range of 3.125–100 µg/mL concentrations. The PIO standard bioanalytical chromatogram appears in Figure 2. The calibration curves were obtained by plotting graphs of the ratio of PIO peak area to internal standard versus PIO concentration for both enantiomers. The mean linear regression equations for PIO-S and R for the concentration range of 3.125–100 µg/mL were determined using the least squares technique, where y = 0.013x + 0.0022, y = 0.0423x − 0.0049 respectively (y is the peak area ratio enantiomer to I.S. and x is the concentration of each enantiomer in plasma). The levels of each enantiomer in plasma were represented by their peak area ratios to the I.S. peak area. With an R^2^ correlation coefficient of 0.9999 for both enantiomers, the calibration curve created during validation demonstrated linearity throughout the standard range investigated. For both isomers, the LOQ obtained was 1.125 µg/mL at a specified signal-to-noise ratio of about 10. At three distinct concentrations (6.25, 25, and 50 µg/mL of PIO racemates in plasma), the absolute recovery for each enantiomer was evaluated. The range of mean percentage recovery for the two enantiomers was 95.3% to 97.38 ± 2.99%. At 10 µg/mL, a mean percentage recovery of 97.80 ± 2.99% was attained for the internal standard.

#### 2.3.2. Accuracy and Precision

For PIO-S, the % CV values of within-run precision increased from 0.38428% to 0.70121%, while the % RE of within-run precision increased from 0.0529% to 0.16%. For PIO-R, the % CV values of within-run precision ranged from 0.16066% to 0.98899%, and the % RE of within-run precision ranged from 0.0642% to 0.2458%. For between-run precision, the % CV values for PIO-S ranged from 0.2080% to 0.7919%, while the % RE varied from 0.0624% to 0.2701%. For PIO-R, the % CV values for between-run precision differed from 0.3972% to 0.7798%, and the % RE varied from −0.3641% to 0.1369%. The precision results at LQC, MQC, and HQC were within a variation of +15% CV, while the precision at LLOQ had a variation of +20% CV. The accuracy results at LQC, MQC, and HQC were within a variation of +15% of the expected concentration. In contrast, LLOQ accuracy had a variation of +20% of the expected concentration (Table 1).

#### 2.3.3. Stability

The stability of PIO enantiomers was examined using benchtop, freeze–thaw, and long-term analyte stability experiments on samples at LQC and HQC. These stability tests and accuracy were discovered to be within ±15% of the anticipated concentration at LQC and HQC, which is consistent with the standards for bioanalytical testing established by the US FDA (Table 2). These findings show that the enantiomers of PIO were stable and did not convert or racemize throughout the investigation. The technique adopted in this study has therefore proven effective for routine analysis.

#### 2.3.4. Recovery

At LQC, MQC, and HQC, respectively, the extraction recoveries for PIO-S from rat plasma were 96.3%, 97.3%, and 96.8%. The extraction recoveries for PIO-R were 99.7%, 99.2%, and 99.9%, respectively, at the same QC levels. Our findings show that both PIO enantiomers may be effectively extracted by utilizing the protein precipitation method with acetonitrile serving as the precipitating agent, yielding consistent recovery rates.

### 2.4. Invitro Assay on Glucose Uptake Study on 3T3-L1 Cell Lines by Flow Cytometry

In this work, 2 test compounds were involved to assess the glucose uptake assay on the cell line chosen, namely 3T3-L1. In the treatment of the cells, the concentrations of the compounds were as follows: Control (No treatment, only PBS), PIO-R and S (6.25, 12.5, 25, 50, 100 µg/mL). In the gated 3T3-L1 singlets, a 2-NBDG histogram identifies the cells at the M1 and M2 phases (M1 refers to the negative expression and M2 to the positive expression/region). M1 and M2 gating can be optimized using software (Cell Quest Software, Version 6.0) analysis. The study considered %Cells expressed in M2. When compared to PBS alone, PIO-R and PIO-S demonstrated dose-dependent glucose absorption for 2-NBDG relative fluorescence intensity. PIO-R showed enhanced glucose absorption compared to PIO-S. In the illustration, 2-NBDG expression was displayed using histograms overlaid on PIO-S and PIO-R treated 3T3-L1 cell lines (Figure 3). A bar graph plotting the percentage of cells expressed by 2-NBDG uptake for control, PIO-R, and PIO-S mixtures treated with 3T3-L1 cells after drug treatment is portrayed in Figure 4, along with the results represented in Table 3 based on one-way *ANOVA* followed by Tukey’s multiple comparison test was employed.

### 2.5. Applications of Pharmacokinetics of Pioglitazone (PIO)

The plasma concentrations of two racemic PIO enantiomers were examined after oral administration of 30 mg/kg PIO racemic to female Albino Wistar rats. Female rats’ mean plasma levels of both PIO enantiomers were plotted against time in Figure 5. Table 4 summarizes the non-compartmental methods-derived pharmacokinetic characteristics of two enantiomers in albino Wistar female rats. (PIO-R and PIO-S was rapidly absorbed from tablets in female rats, with identical Tmax’s of 4 h, but PIO-S was eliminated faster and had a shorter T1/2. At each time point, the concentration plasma profile of (R)-PIO was higher than that of PIO-S in female rats depicted in Figure 5. Compared to PIO-S, (R)-PIOs Cmax, AUC last (h*μg/mL), AUCINF_obs (h*μg/mL), and AUC_% Extrap_obs (%) were about 18.73, 260.56, 266.49 and 9.35 times as high, respectively, indicating improved adsorption and distribution as shown in Table 4 [12]. These findings imply that PIO is disposed of in rats in an enantioselective manner. Using a student’s *t*-test (Graphpad’s Prism v.9 software), we compared the pharmacokinetic parameters of PIO-R enantiomer with those of PIO-S, and the majority of the parameters had a *p* value greater than 0.05. We compared the pharmacokinetic parameters of PIO-R enantiomer with those of PIO-S-enantiomer, and the majority of the parameters had a *p* value greater than 0.05. Results showed considerable differences between R and S-enantiomers, with PIO-R having the highest activity. The observation that the PIO-R enantiomer had higher concentrations in plasma and a lower affinity for PPAR-gamma receptors, as indicated by the AUCisomer (R)/AUCisomer (S) ratio of more than 2.0, suggests that stereoselectivity plays a crucial role in the pharmacological activity of PIO [13]. Stereoselectivity discrepancies between PIO enantiomers in rats may be caused by chiral inversion or a different metabolism rate in rats, and more research is needed to understand the precise causes. 

### 2.6. Estimation of Glucose in Rat Plasma by Semi Automated Biochemistry Analyzer

A semi-automated biochemistry analyzer analyzed kinetic plasma samples. The concentration (μg/mL) against time points (hours) of PIO was plotted to estimate glucose concentration in rat plasma, as displayed in Figure 6. PIO-R decreased glucose (μg/mL) concentration in rat plasma compared to PIO-S, indicating that PIO-R suppressed glucose enhancement (Table 5). GraphPad Prism 9’s one-way *ANOVA* followed by Tukey’s multiple comparison test were used to evaluate PIO R- and S enantiomers glucose concentrations in rat plasma. The results revealed that they were almost a significant difference after 4 h (Figure 6). *p* values were higher than 0.05. Therefore, by using the fixed time kinetic approach, we may conclude that the PIO-R demonstrated increased activity compared to PIO-S in racemic form. However, additional scrutiny is required to validate these observations in human participants.

## 3. Materials and Methods

### 3.1. Chemicals and Reagents

Aurobindo Pharma, Hyderabad provided PIO racemic form and PIO-R and S for bioanalytical studies. The internal standard, Glimepiride, was obtained from Hetero Labs, Hyderabad. Merck, India supplied Ammonium acetate, Acetonitrile, and methanol HPLC grade for the study, while Millipore, MA, USA provided 0.45 µm pore size filters for filtering the mobile phase and solutions. The in vitro study utilized 3T3-L1-mouse embryo fibroblast cell lines (NCCS, Pune, Maharastra, India) and DMEM glucose-free medium (Himedia, Mumbai, India) for cell culture. The researchers also used adjustable multichannel pipettes and Fetal Bovine Serum (#RM10432, of Himedia, Mumbai, India), D-PBS (#TL1006, of Himedia, Mumbai, India), 2-NBDG (Invitrogen: Cat no. 11046, Cayman Chemical, Rajasthan, India), a 6-well cell culture plate (Biolite—Thermo, Bengaluru, Karnataka, India), 50 mL centrifuge tubes (# 546043 TORSON, Mysuru, Karnataka, India). A glucose reagent kit from AGAPPE (Mumbai, India) was used to assess glucose in rat plasma utilizing a semi-automated biochemistry analyzer (Labmate, Ranchi, India). In the study, all chemicals used were over 95% pure.

### 3.2. Animals

The study utilized Albino Wistar female rats (weighing 200–250 g) procured from a CPCSEA-approved commercial breeder called the Center for Experimental Pharmacology and Toxicology. The rats were kept in a sterile laboratory environment and permitted at least one week to acclimatize before the study began. The rats received unlimited water access during the experiment and fasted for 12 h before the treatments were administered. All animal care instructions and treatments followed the regulations suggested by the board of CPCSEA. The animal ethical committee of the JSS Academy of Higher Education & Research reviewed and approved the animal use and care protocol (Reg No: 261/PO/ReBi/S/2000/CPCSEA).

### 3.3. Chromatographic Parameters

A Shimadzu LC-2030C Plus Prominence I High-Performance liquid chromatographic system (HPLC) equipped with a UV detector was used for chromatographic studies. A Phenomenex-manufactured lux i-Amylose-3 (150 mm × 4.6 mm), 5 µm column was utilized for the chromatographic separations. The mobile phase is composed of 10 mM ammonium acetate acetonitrile (60:40, *v*/*v*). Through the binary flow pump, isocratic elution was obtained at a flow maintained at 0.6 mL/min and further the wavelength detection by UV at 265 nm. Throughout the study, column temperature was fixed to 35 °C and adjusted with an injection volume of 20 µL. Data acquisition and integration were evaluated using Lab Solutions software version 5.90.

### 3.4. Standard and Stock Solution Preparations

A 1.0 mg/mL concentration of the primary stock of racemic PIO-R and S, and I.S. (Glimepiride) solutions was obtained by using acetonitrile as a diluting agent. Diluting the principal stock solutions with acetonitrile yielded concentrations ranging from 3.125 to 100 µg/mL for racemic PIO working solutions. Similarly, glimepiride (I.S.) was prepared at a known concentration of 10 µg/mL in acetonitrile. Quality control solutions, including LLOQ at 3.125 µg/mL, LQC at 6.25 µg/mL, MQC at 25 µg/mL, and further HQC at 100 µg/mL, were produced in a way similar to the standard experimental solutions mentioned above.

### 3.5. Sample Preparation

Liquid–liquid extraction was used for sample preparation. The supernatant, i.e., blank plasma (100 µL), was spiked with PIO (200 µL) solution and IS (100 µL) and vortexed for 45 s. Acetonitrile was used to dilute the solution to 1.5 mL, and the finished product was centrifuged at 10,000 rpm for 10 min at 4 °C. Syringe filters were used to filter the final supernatant liquid before it was injected into the HPLC system for analysis.

### 3.6. Non-Radioactive Glucose Uptake Assay of 3T3L1 Cell Lines Utilizing Flow Cytometry

The process involved seeding cells in six-well plates and allowing them to incubate overnight. The medium was replaced later with a serum-free, glucose-free DMEM solution that contained insulin and a fluorescent glucose analog called 2-deoxy-2-[(7-nitro-2,1,3-benzoxadiazol-4-yl) amino]-D-glucose (2-NBDG) [14], followed by further testing samples (50 μM). After a 24-h incubation period, the culture medium was extracted, and the cells were thoroughly washed twice with phosphate-buffered saline (PBS). The cells were detached by trypsin, harvested, and kept at 4 °C in FACS tubes. Cells were centrifuged at 2500 RPM for 5 min at 25 °C before being resuspended in 0.5–1 mL of PBS. The cells were examined using a flow cytometer, which collected fluorescence data from 10,000 single-cell events. FITC (Fluorescein isothiocyanate) signals or intensity were detected in the FL1 (Fluorescence 1) channel, which was intended to detect cell absorption of 2-NBDG’s excitation and emission at 465 and 540 nm. The relative amount of fluorescence intensity (FI) was calculated using BD Cell Quest Pro (Version: 6.0) software by subtracting the background FI from the FI of a single cell’s treatment with or without the addition of the 2-NBDG factor.

### 3.7. Estimation of Glucose in Rat Plasma by Semi Automated Biochemistry Analyzer

The semi-auto biochemistry analyzer used in this study operates on filter photometry. Initially, distilled water was pumped through the machine after turning it on. Each parameter was programmed, and the system was set to flow cell mode with printer mode selected. Estimating glucose levels was carried out using pharmacokinetic plasma samples. Containers were labeled as blanks, standards, and tests. A 1000 µL quantity of standard reagent and 10 µL of distilled water were combined to make a blank analytical tube. A 1000 µL quantity of working reagent and 10 µL of glucose standard were assembled into the sample tube. Finally, 1000 µL of standard reagent and 10 µL of plasma sample were well-mixed in the testing tube. Later, the optical density (T1) was recorded for about 30 s after adding the standard or sample. Similarly, after 60 s of the first reading recording, the second estimation (T2) was measured. The fluorescence intensity of the sample was solely proportional to the concentration of glucose present in the specimen. The concentration of glucose (mg/dL) in the specimen (plasma) was estimated using the fixed time kinetic method, by employing the formula (Absorbance of the sample/Absorbance of the standard) × 100.

### 3.8. Validation of Bioanalytical Methods

#### 3.8.1. Calibration Curve

The concentrations of both enantiomers ranged from 3.125 to 100 µg/mL and were produced by spiking blank plasma samples with 200 µL of PIO racemic working standard solutions and 10 µL of I.S. Samples for calibration were prepared as mentioned above. To establish the calibration curves, linear regression least squares analysis was applied to plot peak area ratios vs. concentrations of enantiomer spikes in samples by using Microsoft^®^ Excel (https://www.microsoft.com/en-us/microsoft-365/excel). Each enantiomer’s detection limit (LOD) and quantification limit (LOQ) were considered.

#### 3.8.2. Recovery

We assessed the peak area ratios of the enantiomer to the internal standard at 3 concentrations (every five runs) based on post-extraction spiked samples to determine enantiomer recovery. Similarly, I.S. recovery was also evaluated at 10 µg/mL [15]. The analytes’ and I.S.’s absolute recoveries do not have to be 100%, but the recovery limit should be constant, unambiguous, and replicable.

#### 3.8.3. Precision and Accuracy

On five consecutive days, five replicates with similar concentrations were compared to test precision and accuracy. Precision was quantified by determining %CV (Coefficient of variation) for all four QC levels [16]. While accuracy was represented as a percentage relative error (R.E.%). Except for the LOQ, the acceptable bounds for precision and accuracy were 15% relative standard deviation and 15% relative error, respectively [17].

#### 3.8.4. Stability

Biomatrix stability was tested for six hours at ambient temperature (25–22 °C) for each enantiomer (bench top). PIO racemate samples were stored at around −20 °C for 30 days to evaluate their freeze stability throughout three stages. Throughout each freeze-thaw cycle, spiked plasma samples were frozen for 24 h at −20 °C and thawed at room temperature. To meet the compliance criteria for stability, the R.S.D. % compared to the recently prepared standard must be within 15%.

### 3.9. Pharmacokinetic Investigations in Rats: A Preliminary Study

We utilized the validated bioanalytical method to pharmacokinetic studies of PIO in rats. For the study, Wistar albino female rats weighing 200–210 g at 4–6 weeks of age were recruited. Animals were acclimatized for 7 days under laboratory conditions. An oral dose of 30 mg/kg of racemic PIO was administered to each rat after overnight fasting with free access to water. After dosing, blood was collected at 0, 1, 2, 4, 6, 8, 12, 24, and 48 h from the respective animals through a retro-orbital puncture in an anti-coagulant (EDTA) containing vial. The collected blood was centrifuged at 3500 rpm for 10 min at 10 °C. The supernatant was injected into the HPLC system. A graph of plasma concentration *v*/*s* time was constructed to obtain various pharmacokinetic parameters using Phoenix WinNonlin 8.1 software. GraphPad Prism 9 software was used to assess the differences in probability values through one-way *ANOVA* followed by Tukey’s multiple comparison test for the pharmacokinetic parameters of PIO enantiomers.

### 3.10. Data Analysis

Phoenix WinNonlin 8.1 software was used to analyze pharmacokinetic parameters. Source data analyzed for maximum plasma concentration (Cmax) and maximal plasma concentration-time (Tmax). All statistical calculations were performed using GraphPad Prism 9 software. Using the DAS outcome, a non-compartmental model with pharmacokinetic parameters was derived. Data were displayed as mean values and standard deviation (S.D.). Based on a significance level of *p* 0.05, the one-way *ANOVA* followed by Tukey’s multiple comparison test was performed to assess the statistical significance of the variations in pharmacokinetic parameters between the two enantiomers.

## 4. Conclusions

Pioglitazone (PIO ) is a chiral drug that is used to treat type 2 diabetes. The two enantiomers of PIO have different pharmacokinetic profiles, meaning that they are absorbed, distributed, metabolized, and eliminated differently by the body. To optimize the therapeutic efficacy of chiral drugs, it may be necessary to separate the two enantiomers and administer only the active isomer. Additionally, the HPLC method developed and validated in this study provides a reliable and sensitive tool for measuring PIO enantiomers in plasma. In 3T3 L1 cells, PIO-R exhibited greater glucose uptake by relative fluorescence intensity for 2-NBDG than PIO-S. This method can be used in future studies aimed at investigating the pharmacokinetics and pharmacodynamics of PIO after oral administration of racemic PIO at 30 mg/kg in female albino Wistar rats. In the case of PIO, the two enantiomers, PIO-R and PIO-S, have different pharmacokinetic and pharmacodynamic properties. Our results showed that PIO-R had higher concentrations in plasma and a lower affinity for PPAR-gamma receptors than (S)-PIO. This suggests that PIO-R may be responsible for the majority of the pharmacological activity of the racemic mixture. The AUCisomer (R)/AUCisomer (S) ratio of more than 2.0 indicates that the concentration of PIO-R is more than two times higher than that of PIO-S, which supports the notion that PIO-R plays a more significant role in the pharmacological activity of PIO. The results of this study highlight the importance of considering enantioselectivity in drug development and clinical practice, as the pharmacokinetics and pharmacodynamics of enantiomers can differ significantly. The discovery of discrepancies in the plasma levels of PIO enantiomers in female rats highlights the importance of considering gender differences in drug metabolism and distribution. This finding suggests that factors such as sex hormones, body composition, and liver enzyme activity may play a vital role in the observed stereoselectivity. Further investigations into the underlying mechanisms of these differences may lead to the development of more targeted and effective treatments for female patients and are needed to identify the precise causes of these stereoselectivity variations. Ultimately, this knowledge could lead to the development of more effective therapies for a range of conditions. Overall, our study provides important insights into the pharmacokinetics of PIO enantiomers in rats and their potential impact on drug efficacy.

## Figures and Tables

**Figure 1 molecules-28-04911-f001:**
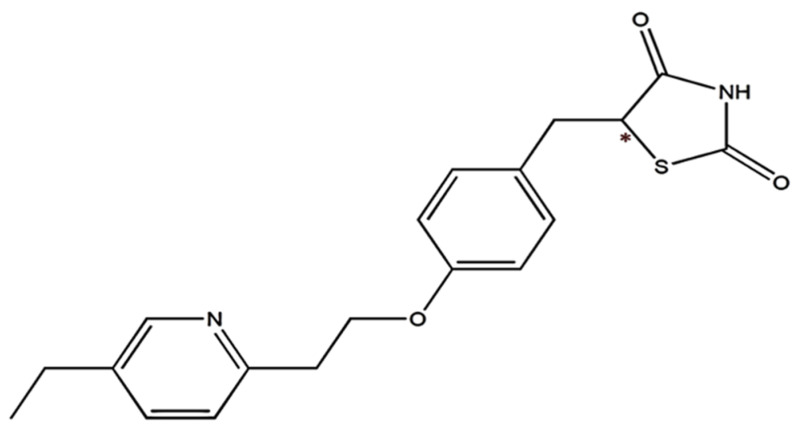
Chemical structure of PIO. *—Denotes the chiral center present in the PIO structure.

**Figure 2 molecules-28-04911-f002:**
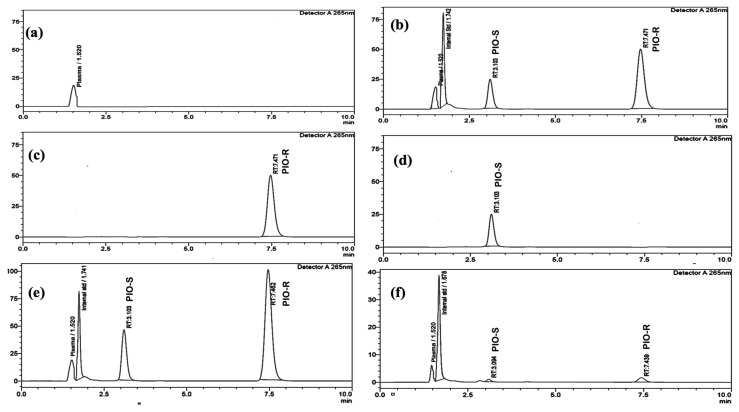
(**a**) Typical chromatograms of rat blank plasma, (**b**) rat blank plasma spiked with PIO racemate standard solution, and the I.S. (10 µg/mL), (**c**) individual standard chromatogram of PIO-R (100 µg/mL) and (**d**) PIO-S (100 µg/mL), and (**e**) plasma sample collected at 4 h of an oral PIO racemate dose of 30 mg/kg, and (**f**) 48 h following an oral PIO racemate dose of 30 mg/kg.

**Figure 3 molecules-28-04911-f003:**
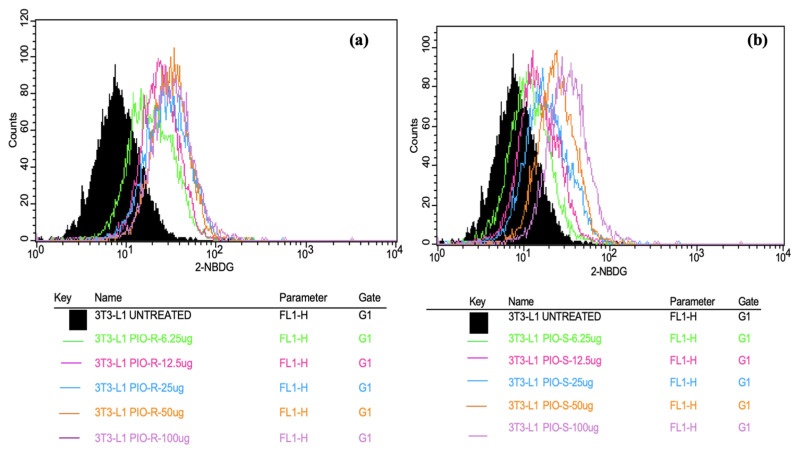
Overlaid histograms representing the 2-NBDG expression observed in PIO-R (**a**) and PIO-S treated with 3T3-L1 cells (**b**).

**Figure 4 molecules-28-04911-f004:**
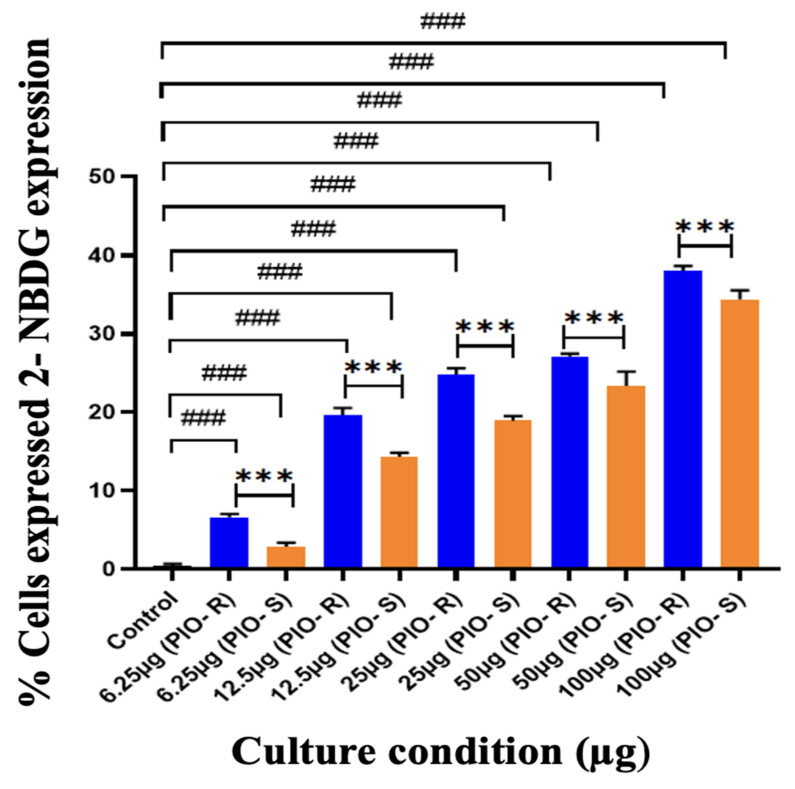
A bar graph plotting the percentage of cells expressed by 2-NBDG uptake in the control, PIO-R, and PIO-S mixtures treated with 3T3-L1 cells after drug treatment. ***—Denotes *p* < 0.001 significant difference in comparison with PIO-R and PIO-S, ###—Denotes *p* < 0.001 significant difference in comparison with the control group. Data represent mean ± SD (*n* = 6) one-way *ANOVA* followed by Tukey’s multiple comparison test was employed.

**Figure 5 molecules-28-04911-f005:**
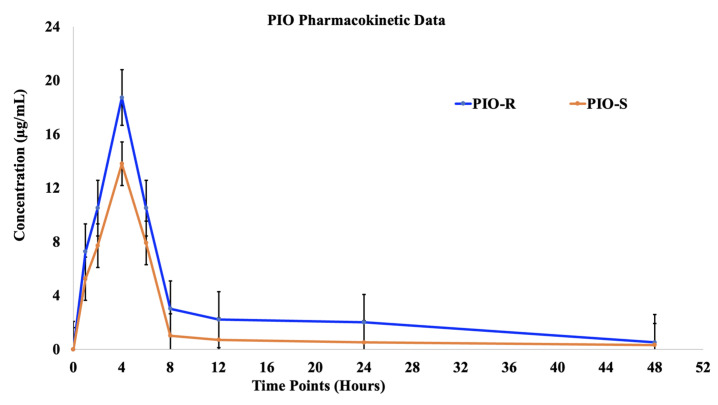
The plasma concentrations (mean ± S.D.) time profile of (R)-PIO and (S)-PIO in female rats after receiving 30 mg/kg oral doses of PIO-R and PIO-S.

**Figure 6 molecules-28-04911-f006:**
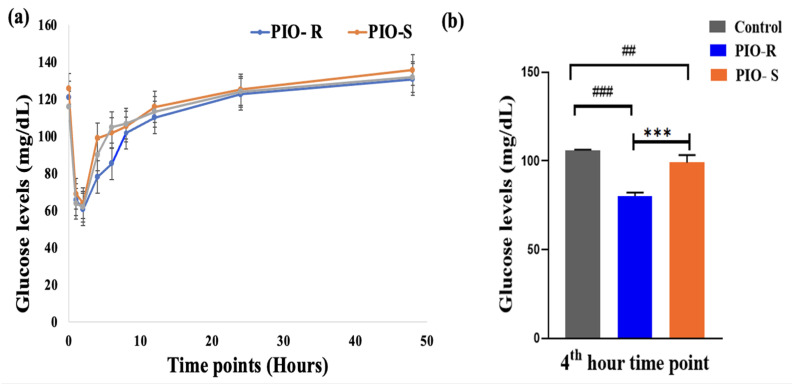
A graphical representation of Control, PIO-R and PIO-S in estimating glucose concentration in rat plasma (**a**), the bar plot represents the significant difference in plasma levels at the 4th-hour plasma interval using GraphPad Prism 9 (**b**). ***—Denotes *p* < 0.001 significant difference in comparision with PIO-R and PIO-S, ### and ##—Denotes *p* < 0.001 and *p* < 0.01 significant difference in comparision with Control group. Data represent mean ± SD (*n* = 6) one-way *ANOVA* followed by Tukey’s multiple comparison test was employed.

**Table 1 molecules-28-04911-t001:** Precision and accuracy results both within and between runs at LLOQ, LQC, MQC, and HQC.

Quality Control Samples	Concentration (µg/mL)	PIO-S	PIO-R
% RE (within runs)			
LLOQ	3.125	0.0843	0.11692
LQC	6.25	0.0630	0.10144
MQC	25	0.16	0.24584
HQC	100	0.0529	0.06424
% RE (between runs)			
LLOQ	3.125	0.2701	0.04246
LQC	6.25	0.0624	0.13696
MQC	25	0.08	−0.36416
HQC	100	0.146	0.02426
% CV (within runs)			
LLOQ	3.125	0.5551	0.16066
LQC	6.25	0.7012	0.71591
MQC	25	0.3842	0.98899
HQC	100	0.5476	0.60702
% CV (between runs)			
LLOQ	3.125	0.7919	0.77982
LQC	6.25	0.7014	0.39726
MQC	25	0.2080	0.47689
HQC	100	0.4585	0.54784

**Table 2 molecules-28-04911-t002:** Results for stability studies.

Quality Control Samples	Conc (µg/mL)	Accuracy (%)		%CV	
		PIO-S	PIO-R	PIO-S	PIO-R
**Bench top stability**					
LQC	6.25	99.91	99.95	0.6471	0.0933
HQC	100	99.76	99.74	0.5394	0.5788
**Freeze–thaw stability**					
LQC	6.25	99.91	99.68	0.6467	0.9271
HQC	100	99.69	99.90	0.7404	0.4730
**Long-term analyte stability**					
LQC	6.25	99.91	99.95	0.6467	0.4625
HQC	100	99.82	100.20	0.5621	0.1524

**Table 3 molecules-28-04911-t003:** The percentage of cells expressed 2-NBDG uptake after drug treatment in Control, PIO-R, and PIO-S treated 3T3-L1 cells.

Concentration	Mean ± SD (*n* = 6) Measurable	
	PIO-R	PIO-S
Control	0.46 ± 0.20	0.46 ± 0.20
6.25 µg	6.65 ± 0.40	2.89 ± 0.49
12.5 µg	19.68 ± 0.88	14.34 ± 0.53
25 µg	24.84 ± 0.77	18.93 ± 0.60
50 µg	27.14 ± 0.31	23.38 ± 1.82
100 µg	38.05 ± 0.58	34.39 ± 1.15

**Table 4 molecules-28-04911-t004:** Pharmacokinetic parameters of PIO enantiomers following oral administration of racemic PIO in female albino Wistar rats (30 mg/kg).

Parameters	Mean ± SD (*n* = 6)	
	PIO-R	PIO-S
Dose (mg/kg b.w.)	15.00	15.00
Cmax (μg/mL)	18.73 ± 3.5	13.80 ± 5.18
Tmax (h)	4.00	4.00
AUClast (h*μg/mL)	260.56 ± 15.36	127.50 ± 4.86
AUCINF_obs (h*μg/mL)	266.99 ± 19.51	133.38 ± 9.03
AUC_%Extrap_obs (%)	9.35 ± 2.71	4.41 ± 0.58
T1/2 (h)	12.47 ± 0.36	12.36 ± 0.55
MRTlast (h)	10.03 ± 0.54	10.74 ± 0.81

**Table 5 molecules-28-04911-t005:** Results of glucose estimation in rat plasma by a semi-automated biochemistry analyzer.

Time Points (Hours)		Glucose Estimation in Plasma Samples Collected at Different Time Points (Mean ± SD) (*n* = 6)	
	Control	PIO R	PIO S
0	115 ± 0.38	120.94 ± 1.09	125.68 ± 1.02
1	63.76 ± 0.58	65.85 ± 1.02	68.99 ± 1.65
2	62.11 ± 0.46	60.71 ± 1.51	63.97 ± 1.77
4	89.91 ± 0.51	78.15 ± 2.02	99.05 ± 4.22
6	104.82 ± 0.41	85.31 ± 1.53	101.91 ± 1.79
8	106.81 ± 0.52	101.7 ± 1.16	105.42 ± 2.58
12	113.31 ± 0.42	110.16 ± 1.56	115.89 ± 1.97
24	124.11 ± 0.55	122.62 ± 1.68	125.12 ± 1.39
48	131.91 ± 0.49	130.69 ± 2.02	135.69 ± 1.27

## Data Availability

The authors confirm that the data supporting the findings of this study are available within the article.

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
