# Peer review of "Implications of Pharmacokinetic Potentials of Pioglitazone Enantiomers in Rat Plasma Mediated through Glucose Uptake Assay"

_molecules, 2023, doi:10.3390/molecules28134911_

Round 1

Reviewer 1 Report

Comments on Molecules-2431594

In this manuscript, Tatineni Spandana et al. reported a LC-UV method of the segregation of pioglitazone enantiomers in rat plasma and successfully applied in a PK study. Additionally, a non-radioactive glucose uptake assay was employed to examine the enantiomers in 3T3-L1 cells by flow cytometry and their potential effects on glucose regulation. The findings are overall interesting and informative to future investigations, while a few points should be addressed properly before being considered for publication.

1. Fig 2A: a small peak seemed to have the same retention time as IS (at 1.7 min) in the blank rat plasma. How many blank plasma samples have been analyzed in the study? What is the average peak intensity of this potential interference?

2. Please include the detailed Stability results.

3. Line 217: “Untreated test compound” should be interpreted as vehicle control.

4. Fig 3B: a reasonable dose dependency could be observed except for Pioglitazone-S at 25 μg/mL, it’s strongly recommended to repeat the assay to rule out technical artifacts. Also, please include a representative snapshot of the gating.

5. Fig 4: the number of replicates (n) and statistical analysis were missing. What might be the reason of relatively high variation in the vehicle controls.

6. Fig 5: the current time points of sample collection may not enable the capture of the full PK profile, please add 3-4 more in the first 12 hours post-dosing.

7. Line 257-259: the authors claimed that R-PIO had a lower affinity for PPAR-gamma receptor as indicated by the ratio of AUC(R)/AUC(S) larger than 2. However, the AUCs listed in Table 3 were quite close between the enantiomers.

8. Table 3: the dose of R-PIO and S-PIO could not be both 30 mg/kg since the rats were dosed with PIO racemic at 30 mg/kg. The data should be interpreted as mean+/-SD. The PK parameters describing the elimination phase such as t1/2 and MRT are better to be obtained from iv instead of po studies.

9. Fig 6: please include the vehicle control group.

10. Line 280-282: the conclusion of R-PIO demonstrated increased activity compared to S-PIO might be debatable since their in vivo exposure were not identical.

11. Please format the figures/tables according to the journal’s standards.

Moderate editing of English language is required.

Author Response

Good Evening, 

we authors express their sincere gratitude to the editorial team and the worthy reviewers for considering our research work for peer review process. Please find below the point-to-point responses of the respective comments (attached as file ). The requisite changes have been made in the revised manuscript and highlighted with yellow color for the distinction.

Reviewer 2 Report

The author studied the pharmacokinetics of R and (S)-Pioglitazone in rat plasma. This article is well written well, and the research is very meaningful. However, some small details need to be revised. Therefore, I recommend accepting it after a minor revision.

The specific revision opinions are as follows:

1.      The author chooses Lux I-Amylose-3 Column (150 x 4.6mm, 5µm), and is there any reference?

2.      Line 164, 1000-10µg/mL is recommended to modify to 10-1000µg/mL.

3.      Figure 6 is not fully displayed, please adjust it. For example, Pio-R and Pio-S.

4.      Line 320, Does the author adopt an equivalent to elution program? In addition, companies that detect instruments should be marked.

Author Response

Good evening, 

We authors express their sincere gratitude to the editorial team and the worthy reviewers for considering our research work for peer review process. Please find below the point-to-point responses of the respective comments. The requisite changes have been made in the revised manuscript and highlighted with yellow color for the distinction.

Round 2

Reviewer 1 Report

I'd thank the authors for the revision work and suggest to accept the paper for publication.

English language is fine.

Author Response

Dear Reviewer , Thank you. 

Regards